# Prevalence of *Helicobacter pylori, Salmonella typhi, Plasmodium falciparum*, and *Toxoplasma gondii* infections and levels of liver function markers among Hepatitis B Virus infected Ghanaians: A cross-sectional study in the Greater Accra Region

Eric NY Nyarko[1]*, Ebenezer K Amakye[2], Emmanuel K Ofori[1], Michael Appiah[3], Manfred Anim[4], Nathaniel L Lartey[5], Samuel Ametepe[6], Evans A Adu[7,8], Justice Kumi[9], Derrick N. D. Dodoo[10], Monica Adom[11], Emmanuel Owiredu[12], Esther Kwafo[13], Christian Obirikorang[7,8]

1 Department of Chemical Pathology, University of Ghana Medical School, College of Health Sciences, University of Ghana, Accra, Greater Accra Region, Ghana, 2 Department of Biomedical Laboratory Sciences, School of Allied Health Sciences, University for Development Studies, Tamale, Northern Region, Ghana, 3 Department of Medical Laboratory Technology, Accra Technical University, Accra, Ghana, 4 Hannover Medical School, Hannover, Germany, 5 Department of Microbiology and Immunology, University of Michigan Medical School, Ann Arbor, Michigan, United States of America, 6 Department of Medical Laboratory Science, Koforidua Technical University, Koforidua, Ghana, 7 Department of Molecular Medicine, School of Medicine and Dentistry, Kwame Nkrumah University of Science and Technology, Kumasi, Ashanti Region, Ghana, 8 Global Health and Infectious Diseases Group, Kumasi Centre for Collaborative Research, Kumasi, Ghana, 9 Noguchi Memorial Institute for Medical Research, College of Health Sciences, University of Ghana, Accra, Ghana, 10 Department of Medical Laboratory Science, Baldwin University College, Accra, Ghana, 11 Department of Medical Microbiology, University of Ghana Medical School, College of Health Sciences, University of Ghana, Accra, Greater Accra Region, Ghana, 12 Department of Biochemistry and Biotechnology, Kwame Nkrumah University of Science and Technology, Kumasi, Ashanti Region, Ghana, 13 Laboratory Department, Impact Medical and Diagnostic Centre, Accra, Ghana

* eynyarko@ug.edu.gh

## Abstract

Coinfection of humans with Hepatitis B Virus (HBV) and non-viral pathogens may worsen the outcome of HBV infection on the liver. This study determined the prevalence of *Heliobacter pylori, Salmonella typhi, Plasmodium falciparum*, and *Toxoplasma gondii* among Hepatitis B Virus (HBV)-infected persons in the Greater Accra Region (GAR) of Ghana and examined how such co-infections might affect the levels of selected liver function markers (LFM). The design was cross-sectional, involving 120 HBsAg-positive HBV-infected persons. Blood samples were collected. *H. pylori, S. typhi, P. falciparum*, and *T. gondii* were screened for, from the blood using lateral flow immunochromatographic assays. *P. falciparum* infection was further confirmed by blood film microscopy. LFM's and blood platelets were measured using clinical chemistry and reflective light suppression/silicon photomultiplier techniques respectively. Data were analyzed using SPSS v. 23.0, and GraphPad 7.0.

**Data availability statement:** All relevant data are within the paper and its Supporting Information files.

**Funding:** The authors received no specific funding for this work.

**Competing interests:** The authors have declared that no competing interests exist.

**Abbreviations:** ALT, Alanine transaminase; AST, Aspartate transaminase; ALP, Alkaline phosphate; BMI, Body mass index; CHRPE, Committee of Human Research, Publication, and Ethics; GAR, Greater Accra Region; GGT, Gamma-glutamyl transferase; HAV, Hepatitis A virus; HBV, Hepatitis B Viral or Hepatitis B Virus; HBsAg, Hepatitis B surface antigen; HCV, Hepatitis C virus; HIV, Human immunodeficiency Virus (HIV); KNUST, Kwame Nkrumah University of Science and Technology; LFM's, Liver function markers; LFT, Liver function tests; RDT, Rapid Diagnostic Test.

Seventy-five,75(62.5%) of the participants were males, and 51.7% were 21-to-30-years old. Prevalence of *H. pylori, S. typhi, T.gondii* and *P. falciparum* were 40.8% (95% CI: 32.5–49.8), 2.5% (95% CI: 0.8–7.1), 19.2% (95% CI: 13.1–27.1), and 2.5% (95% CI: 0.8–7.1) respectively. Levels of Alanine transaminase (ALT) were higher in HBV-H.pylori coinfected persons compared with HBV-only (33.5vs23.5IU,p<0.001). HBV-infected persons in the GAR have high prevalences of *H. pylori* and *T. gondii* coinfections. Some LFM's were elevated due to such coinfections. Care givers would need to widen their screening, monitoring, and diagnosis of HBV coinfections, beyond examination for malaria parasites, and monitor other possible causes of biochemical derangements among HBV-infected persons.

## Introduction

Hepatitis B virus (HBV) infection is reported to account for about 300 million chronic diseases and over 800 thousand deaths each year [1]. HBV infection and their associated diseases continue to ravage Ghanaians, despite the availability of vaccines for prevention [2]. Though the overall prevalence of HBV infections has reduced from over 12% [3] in 2012 to less than 6% in 2022 [4] in the Greater Accra Region (GAR), chronic infections remain in the endemic zone of >8% [5,6]. In the GAR the average prevalence of HBV infection is 5.9%, with prevalence of >9% among health workers [4]. HBV affects the liver, especially among those with genetic vulnerabilities [7], and the hepatitis could be worst among a population like Ghana, where there are high prevalences of other pathogens of hepatic interest. These pathogens include bacteria and protozoa microbes of public health concern and importance, which might further injure the HBV-infected liver, leading to severe liver dysfunction. *Heliobacter pylori* (*H. pylori*) infection has become an important subject of discussion in recent times, due to its association with both gastric and duodenal ulcers [8]. High prevalences have been reported among adult Ghanaians with dyspepsia, and in rural children. Specifically, >75% of Ghanaians with dyspepsia [9], and >14% of Ghanaian rural children are infected with the bacteria [10]. A study in 2024, by Maiorana and colleagues, concluded that *H. pylori* contributes significantly to liver injury and fibrosis in patients with existing liver pathology [11], while a study by Wang *et al.* have indicated concordance of high rates of *H. pylori* infection among Chinese with HBV infection [12].

*Salmonella enterica* serovar typhi (*S. typhi*) affects 1 in every 500 Ghanaians annually [13]. It is the main etiological agent of typhoid or enteric fever in Ghana, and often transmitted via contaminated water and food. Generally, insanitary environments are the main culprits of *S. typhi* transmission and infection. High levels of mono-infections of *S. typhi* have been reported in the Western and Upper West Regions of Ghana [14]. Prevalence of between 1.9% and 5% was reported in the GAR in 2023 [15]. It is important to note, that, in the settings where these studies were carried out, insanitary conditions, inaccurate diagnosis and monitoring techniques – leading to increased chronic infections, with its associated gastrointestinal

complications were present [13–15]. Moreover, studies and publications of *S. typhi* infections among HBV infected persons, or its effects on liver function markers (LFM's) have not been reported in Ghana.

Falciparum malaria is endemic in Ghana, and it's caused by *Plasmodium falciparum*. *Plasmodium falciparum* malaria prevails in 11% of the general Ghanaian population, - data often skewed up due to high infection rates among pregnant women and children under 5-years. It affects 34% pregnant women annually, and kills about 30% of children under 5-years of age [16–18]. The studies which have previously reported HBV-P.falciparum coinfections are that of Asantewaa *et al*., and Helegbe *et al.,* but these were among pregnant women in the Gonja District and the Tamale Municipality respectively [19,20]. It is important to note that, both HBV and *P. falciparum* have part of their lifecycles within the liver, and may have different or combined liver biochemical manifestations, which have not yet been reported in the Ghanaian population, though both infections are endemic.

*Toxoplasma gondii* is a unicellular protozoa parasite which causes Toxoplasmosis. *Toxoplasma gondii* infection is attributable to unhygienic environments [21–23]. In Ghana, *T. gondii* screening is often done for at-risk pregnant women [21], to help manage and protect susceptible fetuses. Moreover, HBV is a major cause of chronic liver disease, and liver dysfunction. Alone, HBV can cause liver cirrhosis and liver cancer. However, the presence, prevalence, and the effect of other organisms such as *H. pylori, S. typhi, P. falciparum* and *T. gondii*, among HBV infected persons have not been fully explored, at least not in recent times, among HBV-infected Ghanaians living in the GAR. As stated above, these bacteria and protozoa parasites are of very high prevalence among Ghanaians, as mono infections. The objective of this study was to find the prevalence of *H. pylori, S. typhi, P. falciparum* and *T. gondii* in persons with HBV infections in the GAR of Ghana, and to examine how such co-infections affect the levels of some liver function markers (LFMs). The findings of this study will provide knowledge and data, which can help to manage HBV infections among the study population, in terms of screening and monitoring of HBV coinfections, interpretation of liver function tests and management of HBV patients.

## Methodology

### Ethics statement

Ethical approval was obtained from the Kwame Nkrumah University of Science and Technology's, (KNUST), Committee of Human Research, Publication and Ethics (CHRPE) (CHRPE/AP/426/19), after submitting written informed consent documents and health facilities approval documents. Written informed consent was obtained from all participants before they were allowed to participate in the study the study.

### Study design, population and case definition

The study design was cross-sectional. Patient's data and blood sample collection were done at the Mamprobi Polyclinic, Ussher Polyclinic, and Kaneshie Polyclinic, all in the GAR of Ghana, between August 2019 and June 2020. In all, 151 patients who tested positive to HBsAg were initially included, but 31 persons were further excluded (see *Supplementary file,* **S1** File) including HBV -HBsAg positive participants, those who tested positive to either Human immunodeficiency Virus (HIV), Hepatitis C virus (HCV), or Hepatitis A virus (HAV) were excluded. Also, those who had received pharmacological treatment recently (≤ 3 months) for any of the investigating non-viral coinfections, and pregnant women were excluded (**S1 File**). The pregnant women were excluded due to hypervolemia, haemodilution and gestational ALP effects on liver function tests, caused by pregnancy states. The study was ethically reviewed and approved by the Committee of Human Research, Publication, and Ethics (CHRPE), of the Kwame Nkrumah University of Science and Technology (KNUST)

### Sample size estimation

The sample size was estimated using the formular, [24]. With a current prevalence (8.36%) of HBV infection among Ghanaian adult population [5], a standard normal variate (z) of 5% (type 1 error, p<0.05), precision (d) of 5%, and proportion of

HBV infected persons in the population, based on the previous study (8.36%), the minimum sample size was estimated as explained by Charan and Biswa (2013) [24] to be 118 participants. However, 151 participants with HBV infection (by HBsAg positivity) were initially recruited into the study, and after further exclusions - explained under population and case definitions, the number ended at 120. HBV infection prevalence was used for the sample size calculation, because the primary population of interest was HBV infected persons. Using the HBV group as the basis of the calculation also contributed to obtaining the right study power to the sample size, while achieving the study objectives in the face of scarce resources.

## Data collection

Structured questionnaire was administered to each consented participant to obtain responses with regards to sociodemographic, and HBV experiences and knowledge. Selected LFM's, and platelets were measured. Patient confidentiality was upheld for all the participants throughout the study. Only the research team and the CHRPE had access to the data, but none of the study authors could identify individual participants during or after data collection.

### Collection and preparation of blood sample for analysis

Six (6) milliliters (ml) of venous blood were collected from each participant into two (2) tubes- 3 ml into serum separator gel tube (SST) and 3 ml into $K_2$EDTA tube. The sample in the SST was allowed to clot (protected from light and heat) at laboratory room temperature for 35–45 minutes and centrifuged at 2500g for 7minutes to obtain sera. The sera were aliquoted into two (2) Eppendorf tubes and stored between -16 and -20° C till assayed.

### Screening of H. Pylori, S. typhi, P. falciparum, and T. gondii

The presence of *H. Pylori, S. typhi* and *T. gondii* in the patient samples were detected following the manufacturer's protocols, respectively, after testing ELISA confirmed controls previously obtained from the Noguchi Memorial Institute for Medical Research (NMIMR), Legon, Accra. Briefly, *H. pylori* IgG/IgM Ab combo cassette (CTK Biotech, Inc., California, USA) based on double-antigen sandwich principle of the lateral flow chromatographic immunoassay was used to screen for *H. pylori*. Aria Typhoid IgM and IgG rapid test kit was used to screen for the presence of *S. typhi* based on the lateral flow immunoassay principle (CTKBiotech Inc., San Diego, USA). *T. gondii* IgG/IgM antibodies were detected based on the lateral flow immunochromatographic technique, using the Accu-Tell Toxo IgG/IgM Rapid test Cassette (Accubiotech, Beijing, China). 5 µl of the $K_2$EDTA-anticoagulated whole blood sample was screened for *P. falciparum* using SD Bioline Malaria Ag P.f (HRP2/pLDH) based on the immunochromatographic lateral flow assay (Standard Diagnostics Inc., Korea). The *P. falciparum* positive samples were confirmed by a trained microscopic examination of thin and thick film prepared for blood smear. Both microscopic and immunochromatographic assays were used for the *P. falciparum* screening to cater for challenges of sensitivity and specificity, respectively associated with each technique.

### Measurement of liver function markers and blood platelets

Autocal (13160) and HumaTrol (13511) calibrated HumaStar 200 Clinical chemistry analyzer (Human Diagnostics worldwide, Germany) was used to spectrophotometrically measure selected liver function markers, based on Beer Lambert's, Kinetic end point and Coupled enzyme assay principles. The platelets were counted from the EDTA whole blood using Mindray 5-parts hematology analyzer based on the reflective light suppression and photomultiplier technologies (Mindray, Shenzhen-China, 2013). AST-to-ALT ratio and Fibrosis-4 (FIB-4) liver fibrotic indices were computed as described previously [25]

## Data analysis

Data collected were analyzed using GraphPad Prism 7, and the Statistical Package for the Social Sciences (SPSS) version 23.0 [26]. The sociodemographic characteristics were analyzed and reported as frequencies and percentages. The

prevalences of *H. pylori*, *S. typhi*, *P. falciparum* and *T. gondii*, among the HBsAg-positive HBV persons were reported with simple bar graph. Continuous LFM's were represented by mean ± standard deviations [Median (IQR)] and were compared between the various coinfections and the HBV only – positive persons without the coinfections, using the independent t-test. P values < 0.05 were deemed statistically significant.

## Results

Table 1 shows the sociodemographic characteristics of the study participants. One hundred and twenty (120) HBV infected persons were in this study. There were more males, 75 (62.5%) than females. More than half (51.7%) of the HBV infected persons were between the ages of 31 and 40 years. Forty 40, (33.3%) of the participants originally hailed from the Greater Accra region, while 1 each (0.8%) was from the Ahafo, Bono and Upper West Regions, respectively. With regards to ethnic distribution, 54 (45%) were Akan, there were 2 (1.7) each of Bissa, Guan and Grussi, but there was no Hausa. Two, 2(1.7%) of the 120 participants were (formally) uneducated, while more than half of them (that is 67, repre-senting 55.8%) had secondary education. Fourteen, 14 (11.7%) were single, 65 (54.2%) were married and the rest were previously married (either divorced or widowed). On the knowledge of other family members who may have HBV infection, 1 (0.8%) patient mentioned the spouse is infected and 111(92.5%) said they didn't know the infection status of any family member (Table 1).

**Table 1. Sociodemographic and anthropometric characteristics of the study participants.**

| Variable | Frequency | Percentage (%) |
|---|---|---|
| *Gender* | | |
| Male | 75 | 62.5 |
| Female | 45 | 37.5 |
| *Age (Years)* | | |
| ≤ 20 | 5 | 4.2 |
| 21- 30 | 62 | 51.7 |
| 31- 40 | 38 | 31.7 |
| 41 - 50 | 14 | 11.7 |
| >50 | 1 | 0.8 |
| *Region of Origin* | | |
| Eastern | 21 | 17.5 |
| Greater Accra | 40 | 33.4 |
| Northern | 4 | 3.3 |
| Oti | 3 | 2.5 |
| Ahafo | 2 | 1.7 |
| Volta | 4 | 3.3 |
| Ashanti | 6 | 5.0 |
| Bono East | 4 | 3.3 |
| Bono | 1 | 0.8 |
| Ahafo | 1 | 0.8 |
| Central | 5 | 4.2 |
| Savannah | 7 | 5.8 |
| Upper East | 9 | 7.5 |
| Upper West | 1 | 0.8 |
| Western | 4 | 3.3 |
| Western North | 8 | 6.7 |

*(Continued)*

Table 1. (Continued)

| Variable | Frequency | Percentage (%) |
|---|---|---|
| *Ethnic Group* | | |
| Akan | 54 | 45.0 |
| Ewe | 9 | 7.5 |
| Ga-Dangme | 29 | 24.2 |
| Grusi | 2 | 1.7 |
| Guan | 2 | 1.7 |
| Gurma | 4 | 3.3 |
| Bissa | 2 | 1.6 |
| Mole-Dagbani | 18 | 14.9 |
| HFE | | |
| Uneducated | 2 | 1.7 |
| Primary | 24 | 20.0 |
| Secondary | 67 | 55.8 |
| Tertiary | 27 | 22.5 |
| *Marital Status* | | |
| Previously Married | 41 | 34.2 |
| Married | 65 | 54.2 |
| Single | 14 | 11.7 |
| *FM with HBV Positive* | | |
| Spouse | 1 | 0.8 |
| Sibling | 3 | 2.5 |
| Not that I know of | 111 | 92.5 |
| None | 5 | 4.2 |

Sociodemographic characteristics were presented as frequency (percentage, %). G. Accra: Greater Accra, HFE: Highest Formal Education, FM: Family Member, HBV: Hepatitis B Virus.

Prevalence of H. pylori, S. typhi, T.gondii and P. falciparum were 40.8% (95% CI: 32.5–49.8), 2.5% (95% CI: 0.8–7.1), 19.2% (95% CI: 13.1–27.1), and 2.5% (95% CI: 0.8–7.1) respectively. This represents 49/120, 3/120, 23/120 and 3/120 of these coinfections respectively (**Fig 1**). With regards to the effect of these HBV-coinfections on LFM's, ALT was significantly higher in HBV-H. pylori coinfected persons compared with HBV-only persons (*p < 0.001*). There were no statistically significant differences in the levels of the other markers between the two groups. On the other hand, total bilirubin, total protein, and ALT were higher in HBV-S. typhi coinfected persons compared with HBV-only patients. The levels of direct bilirubin and albumin were higher in the HBV-S.typhi coinfected persons compared with the HBV-only persons, though in both cases, statistical significance cannot be inferred due to the low numbers of the HBV-S.typhi group **Table 2**).

There were no significant differences in the levels of the LFM's nor the fibrotic indices between the HBV-T. gondii coinfected patients and the HBV-only infected patients. Without considering the statistical significance differences in the levels of LFM's between the HBV only and the HBV-P. falciparum, due to the small number of HBV-P. Falciparum infected group, the levels of total bilirubin, indirect bilirubin, albumin and ALT were higher in the HBV-P. falciparum coinfected than those with HBV-only infection (**Table 3**).

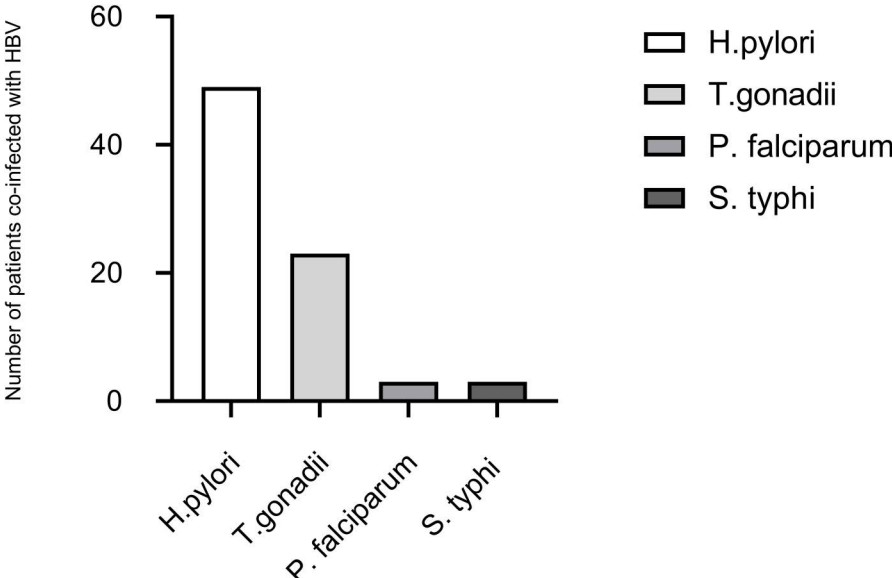

**Fig 1. Prevalence of H. pylori, S. typhi, T. gondii, and P. falciparum among HBsAg positive participants.** The incidence of H. pylori infection in HBsAg $^+$ HBV patients was higher, 49/120 (40.8%), followed by T. gonadii, 23/120 (19.2%) while prevalence of HBsAg $^+$ HBV/ S. typhi co-infection, and HBsAg + HBV/P. falciparum co-infection were 3/120 (2.5%) respectively.

**Table 2. Effects of bacteria pathogen (H. pylori and S. typhi) coinfections on liver function markers of HBV (HBsAg+) persons.**

| Liver Function Markers | HBV only (n= 42) Mean (±SD) Median (IQR) | HBV -H. Pylori (n = 49) Mean (±SD) Median (IQR) | P-value |
|---|---|---|---|
| Total Bilirubin (μmol/L) | 13.5(5.2) 13.5 (10.4 - 16.6) | 15.2 (6.9) 14.6 (10.9 - 18.3) | 0.31 |
| Direct Bilirubin (μmol/L) | 6.1 (3.2) 5.7 (3.5 - 7.9) | 6.3 (3.0) 6.4 (4.25 - 8.55) | 0.98 |
| Indirect Bilirubin, calc. | 7.3 (4.2) 6.7 (3.55 - 9.85) | 8.9 (5.8) 8.1 (4.7 - 11.5) | 0.23 |
| Total Protein (g/L) | 71.5 (8.2) 71.7 (67.2 - 76.2) | 71.3 (9.7) 70.9 (65.25 - 76.55) | 0.91 |
| Albumin (g/L) | 40.0 (6.3) 38.6 (34.1 - 43.1) | 39.8 (8.0) 38.4 (33.55 - 43.25) | 0.94 |
| Globulins, calc. | 31.5 (7.6) 31.6 (27.65 - 35.55) | 31.5 (8.9) 31.2 (26.5 - 35.9) | 0.96 |
| ALT (U/L) | 23.5 (10.1) 21.0 (15.0 - 27.0) | *33.5 (15.1)* 31.8 (20.55 - 43.05) | **<0.001** |
| AST (U/L) | 27.8 (22.1) 26.20 (20.45 - 31.95) | 34.8 (18.8) 33.2 (25.2 - 41.2) | 0.07 |
| γGT (U/L) | 31.7 (13.4) 32.0 (26.0 - 38.0) | 29.7 (10.7) 30.0 (23.0 - 37.0) | 0.25 |
| ALP (U/L) | 161.1 (51.8) 152 (116 - 188) | 152.0 (56.0) 144 (105 - 183) | 0.32 |
| Platelets x 10$^9$/L | 197.4 (49.5) 190 (155 - 225) | 191.9 (54.5) 191 (156 - 226) | 0.44 |
| AST/ALT ratio | 1.35 (1.39) 1.030 (0.785 - 1.275) | 1.28 (1.13) 1.000 (0.675 - 1.325) | 0.71 |

*(Continued)*

**Table 2.** (Continued)

| Liver Function Markers | HBV only (n= 42) Mean (±SD) Median (IQR) | HBV -H. Pylori (n = 49) Mean (±SD) Median (IQR) | P-value |
|---|---|---|---|
| FIB-4 | 0.92 (0.60) 0.85 (0.54 - 1.16) | 1.16 (1.12) 0.930 (0.625 - 1.235) | 0.21 |
| | **HBV only (n= 42)** | **HBV -S. typhi (n = 3)** | **P-value** |
| Total Bilirubin (µmol/L) | 13.5 (5.2) 13.5 (10.4 - 16.6) | 23.6 (8.5) 25.9 (25.9 - 25.9) | |
| Direct Bilirubin (µmol/L) | 6.1 (3.2) 5.7 (3.5 - 7.9) | 12.9 (5.9) 12.1 (12.1 - 12.1) | |
| Indirect Bilirubin,calc. | 7.3 (4.2) 6.7 (3.55 - 9.85) | 10.7 (7.6) 8.5 (8.5 - 8.5) | |
| Total Protein (g/L) | 71.5 (8.2) 71.7 (67.2 - 76.2) | 84.2 (5.0) 84.5 (84.5 - 84.5) | |
| Albumin (g/L) Globulins, calc. | 40.0 (6.3) 38.6 (34.1 - 43.1) 31.5 (7.6) 31.6 (27.65 - 35.55) | 53.2 (2.4) 54.0 (54.0 - 54.0) 30.8 (6.8) 29.2 (29.2 - 29.2) | |
| ALT (U/L) | 23.5 (10.1) 21.0 (15.0 - 27.0) | 43.7 (19.3) 50.0 (50.0 - 50.0) | |
| AST (U/L) | 27.8 (22.1) 26.20 (20.45 - 31.95) | 30.7 (6.7) 29 (29 - 29) | |
| γGT (U/L) | 31.7 (13.4) 32.0 (26.0 - 38.0) | 38.7 (27.7) 35 35 (35 - 35) | |
| ALP (U/L) | 161.1 (51.8) 152 (116 - 188) | 201 (72.0) 219 (219 - 219) | |
| Platelets x10^9/L | 197.4 (49.5) 190 (155 - 225) | 184.7 (62.3) 167 (167 - 167) | |
| AST/ALT ratio | 1.35 (1.39) 1.030 (0.785 - 1.275) | 0.83 (0.45) 0.86 (0.86 - 0.86) | |
| FIB-4 | 0.92 (0.60) 0.85 (0.54 - 1.16) | 0.60 (0.18) 0.69 (0.69 - 0.69) | |

Data is presented as mean±SD. *=p-values< 0.05, **=p-values< 0.01 were considered statistically significant. IQR – Interquartile range, ALT – Alanine transaminase, AST – Aspartate transaminase. GGT - Gamma-glutamyl transferase, ALP – Alkaline phosphatase, FIB-4 - fibrosis 4 indices.

**Table 3. Effects of protozoa parasites (*T. gondii and P. falciparum*) on coinfections on liver function markers of HBV (HBsAg+) persons.**

| Liver Function Markers | HBV only (n= 42) Mean (±sd) | HBV -T. gondii (n = 23) | P-value |
|---|---|---|---|
| Total Bilirubin (µmol/L) | 13.5(5.2) 13.5 (10.4 - 16.6) | 15.2 (7.2) 14.8 (9.35 - 20.25) | 0.26 |
| Direct Bilirubin (µmol/L) | 6.1 (3.2) 5.7 (3.5 - 7.9) | 5.8 (2.6) 6.1 (4.15 - 8.05) | 0.76 |
| Indirect Bilirubin, calc. | 7.3 (4.2) 6.7 (3.55 - 9.85) | 9.3 (5.9) 8.2 (4.2 - 12.2) | 0.23 |
| Total Protein (g/L) | 71.5 (8.2) 71.7 (67.2 - 76.2) | 68.9 (9.4) 70.4 (62.4 - 78.4) | 0.12 |
| Albumin (g/l) | 40.0 (6.3) 38.6 (34.1 - 43.1) | 38.2 (6.9) 37.3 (31.45 - 43.15) | 0.24 |

*(Continued)*

**Table 3.** (Continued)

| Liver Function Markers | HBV only (n= 42) Mean (±sd) | HBV -T. gondii (n = 23) | P-value |
|---|---|---|---|
| Globulins, calc. ALT (U/L) | 31.5 (7.6) 31.6 (27.65 - 35.55) 23.5 (10.1) 21.0 (15.0 - 27.0) | 31.7 (7.9) 29.7 (25.35 - 34.05) 27.3 (14.6) 23.0 (12.75 - 33.25) | 0.75 0.16 |
| AST (U/L) | 27.8 (22.1) 26.20 (20.45 - 31.95) | 29.0 (12.3) 28.4 (18.8 - 38.0) | 0.83 |
| γGT (U/L) | 31.7 (13.4) 32.0 (26.0 - 38.0) | 29.7 (10.8) 26.5 (20.0 - 33.0) | 0.69 |
| ALP (U/L) | 161.1 (51.8) 152 (116 - 188) | 153 (52) 150 (117 - 183) | 0.71 |
| Platelets x10^9/L | 197.4 (49.5) 190 (155 - 225) | 203 (50) 215 (177 - 253) | 0.54 |
| AST/ALT ratio | 1.35 (1.39) 1.030 (0.785 - 1.275) | 1.24 (0.60) 1.14 (0.79 - 1.49) | 0.64 |
| FIB-4 | 0.92 (0.60) 0.85 (0.54 - 1.16 | 0.94 (0.38) 0.96 (0.72 - 1.20) | 0.87 |
| | HBV only (n= 42) | HBV -P. falciparum (n =3) | |
| Total Bilirubin (μmol/L) | 13.5(5.2) 13.5 (10.4 - 16.6) | 26.7 (6.0) 28.4 (28.4 - 28.4) | |
| Direct Bilirubin (μmol/L) | 6.1 (3.2) 5.7 (3.5 - 7.9) | 7.3 (3.4) 6.6 (6.6 - 6.6) | |
| Indirect Bilirubin, calc. | 7.3 (4.2) 6.7 (3.55 - 9.85) | 19.4 (5.0) 20.0 (20.0, 20.0 | |
| Total Protein (g/L) | 71.5 (8.2) 71.7 (67.2 - 76.2) | 74.3 (15.2) 81.0 (81.0, 81.0) | |
| Albumin (g/L) | 40.0 (6.3) 38.6 (34.1 - 43.1) | 50.5 (14.5) 50.7 (50.7, 50.7) | |
| Globulins, calc. | 31.5 (7.6) 31.6 (27.65 - 35.55) | 23.7 (9.5) 20.9 (20.9 - 20.9) | |
| ALT (U/L) | 23.5 (10.1) 21.0 (15.0 - 27.0) | 39.3 (15.1) 43.0 (43.0 - 43.0) | |
| AST (U/L) | 27.8 (22.1) 26.20 (20.45 - 31.95) | 36.3 (7.6) 38.0 (38.0 - 38.0) | |
| γGT (U/L) | 31.7 (13.4) 32.0 (26.0 - 38.0) | 28.0 (4.6) 27.0 (27.0 - 27.0) | |
| ALP (U/L) | 161.1 (51.8) 152 (116 - 188) | 179.7 (54.7) 162.0 (162.0 - 162.0) | |
| Platelets x10^9/L | 197.4 (49.5) 190 (155 - 225) | 201.7 (50.6) 198 (198.0 - 198.0) | |
| AST/ALT ratio | 1.35 (1.39) 1.030 (0.785 - 1.275) | 0.99 (0.26) 0.93 (0.93 - 0.93) | |
| FIB-4 | 0.92 (0.60) 0.85 (0.54 - 1.16 | 0.90 (0.25) 0.85 (0.85 - 0.85) | |

Data is presented as mean (± SD). *=p-values< 0.05, **=p-values< 0.01 were considered statistically significant. IQR – Interquartile range, ALT – Alanine transaminase, AST – Aspartate transaminase. GGT - Gamma-glutamyl transferase, ALP – Alkaline phosphatase, FIB-4 - fibrosis 4 indices.

## Discussions

Our current study examined the prevalence of *Helicobacter pylori, Salmonella typhi, Plasmodium falciparum,* and *Toxoplasma gondii* among HBsAg -positive; Hepatitis B Virus (HBV) infected persons, and how these bacteria or protozoa co-infections might affect the levels of LFM's. The prevalence of *H. pylori* infection in this current study was 40.8%. Studies on the prevalence of *H. pylori* infections in the general (apparently healthy) Ghanaian population was not found, however, prevalences of 75.4% and 14. 2% have been reported among adult Ghanaians with dyspepsia, and among rural Ghanaian children respectively [9,10]. In other countries, such as China, where HBV infections are equally endemic, *H. Pylori* prevalence is also very high. This was reported in a review by Wang et al [12]. Reasons for this concordance between high HBV prevalence and *H. pylori* prevalence is partly due to the life cycle of *H. pylori* through the enterohepatic and biliary system. Within the liver, *H. pylori* can inflame and initiate cytopathic effects on hepatocytes, thereby increasing the liver susceptibility to HBV infection, and injury. Also, in patients with Chronic hepatitis B infection, the presence of other bacterial infections causes more damage on liver cells [27]. Besides, strong associations between *H. pylori* infections and other infectious liver diseases have been reported [12,28]. The injury, inflammatory and cytopathic effects of H. pylori – HBV coinfections, may be observed in the elevation of some transaminases. In our current study, ALT was significantly elevated in H. pylori-HBV coinfections, compared with HBV-only infections. In a previous study, unexplainable elevated transaminases have been observed in individuals with *H. pylori* only infections, and the treatment of it (i.e., the *H. pylori* infection) resulted in decreased levels of such transaminases [29]. Moreover, other studies have found dysregulated LFM's and other biochemical indices that are partly metabolized by the gastrointestinal tract (GIT) and the liver, among persons infected with HBV and *H.pylori* (8, 9).

The prevalence of *Salmonella typhi* (and paratyphi) infections in our current study is approximately 2.5%. No study on the prevalence of *S. typhi* infection among HBV infected persons in Ghana or the GAR was found. Again data on the prevalence and incidence of *S. typhi* and paratyphi infections are scanty, however, according to the Ghana coalition against Typhoid and paratyphoid, typhoid fever affects more than 200 persons per 100,000 Ghanaians every year [13]. Out of this, 1 of every 100 persons die directly from the infection, not counting the associated morbidities [13]. *Salmonella typhi*, is a chronic multisystemic infection, which affects several organs in the body, especially the GIT and the liver. The very low or 'insufficient' prevalence of HBV-S. typhi co-infection in this study, warrants that statistical significance from comparing its LFMs with those having HBV-only infections will be a misinterpretation. Moreover, elevated ALT, conjugated or direct bilirubin and dysregulation of other LFMs have been reported by Pramoolsinsap *et al*., [30] and Ahmed and Ahmed [31]. Reasons for such biochemical manifestations include the endotoxin production and hepato-inflammatory actions of *Salmonella typhi* on the liver [30–32]. Decline and normalization of hepatic transaminases was previously found to be indirectly proportional to increase in albumin levels in a cholestatic hepatitis-typhoid fever patient by Eranda and his colleagues, among a Sri Lanka population [32]. However, a study in Cameroon found reduced S-albumin levels in *S. typhi* patients [33]. A reason for the increased levels of the albumin in HBV-S.typhi co-infection could be attributed to the liver's response to the free radicals generated by *S. typhi* in chronicity, to provide an antioxidative function [34], since albumin is a metabolic antioxidant.

The prevalence of *T. gondii* infection among the HBV patients in our current study is 19.2%. With a wide range of routes of contamination and infection, *T. gondii* prevalence have been reported among different populations in Ghana. Among pregnant women, prevalence between 10% and 60% has been reported based on the age, the type of trade, region of domicile, and importantly the method used for screening or diagnosis [21,22]. Though no study have yet reported the incidence or prevalence of *T. gondii* in HBV infected persons, prevalence of 74.4% have been reported among HIV patients in the central region of Ghana [23]. The common attributable risk factor to high *T. gondii* infections in all these populations are reduced immune systems and poor sanitary conditions [21]– [23]. The levels of all the LFM's and the fibrotic indices were statistically the same for the HBV-T. gondii coinfected persons, and the HB- only infected persons, in this study.

However, other studies have reported that *T. gondii* can be inflammatory and cytopathic, characterized by derangement in LFM's - including the elevation of transaminases [35,36]. These differences between our study, and that of Yahia *et al.*, [35] and Babekir *et al.*, [36] could be attributable to the absence of any coinfection agents, such as HBV in their studies, and the longitudinal nature of Babekir *et al.*'s study with a very large sample size of over 3,000.

In Ghana, malaria, notably falciparum malaria is endemic, especially among infants, children and pregnant women. It makes up over 30% of all outpatient cases and prevails in over 11% of the general Ghanaian population. It has been reported in 34% of pregnant women and expectant mothers, and kills over 30% of children under 5 years [16]– [18]. However, studies on prevalence of malaria among HBV infected (non-pregnant women or children) persons was not found. Our study found approximately 2.5% prevalence of *P. falciparum* in HBV infected people. HBV-*P. falciparum* coinfections have been reported among pregnant women and other groups in the Ghanaian population [19,20]. Literature comparing levels of LFMs in HBV-P. falciparum co-infcetion and HBV-only infections were not found, however, a review involving the comparison of LFM's between HBV-P. falciparum persons and *P. falciparum* only persons found no significant differences in the levels of the total bilirubin [37]. Besides, elevated total bilirubin, unconjugated (indirect) bilirubin, and ALT in HBV-P.falciparum co-infections could be due to accelerated haemolysis of falciparum parasitized cells, and the concomitant inflammation of the liver of HBV infected hepatocytes [38]. These findings indicate the need to screen for some of these protozoa and bacteria parasites, in the management of HBV-infected persons.

## Conclusion

We have demonstrated in this study, that, HBV infected persons in the GAR of Ghana have high prevalences of *H. pylori* and *T. gondii* coinfections. Within the limitations of our sample size(s), these findings require that care givers widen their screening, monitoring, and diagnosis of HBV coinfections, beyond examination for malaria parasites. Also, the level of ALT was significantly elevated in HBV-H. pylori- coinfected persons compared with HBV-only persons. Total bilirubin, direct bilirubin, total protein, albumin and ALT were higher in HBV-S. typhi coinfected persons, compared with HBV-only. The elevation of some LFM's also re-emphasizes the need to screen and monitor other possible causes of biochemical derangements among HBV patients.

## Limitations

1. The application of the findings from the comparison of the LFM's (especially among the HBV-P. falciparum verses HBV only; and HBV-S. typhi verses HBV only*), should be taken with caution since the sample sizes are very small.

2. The study was done in only the GAR of Ghana, as such, applications of the findings in the other regions or the entire Ghanaian population should be done with caution, as the epidemiology of HBV infections in these regions might differ.

## Supporting information

**S1 File. Flow chart.**
(DOCX)

**S2 File. STROBE checklist.**
(DOC)

## Author contributions

**Conceptualization:** Eric NY Nyarko, Ebenezer Amakye, Christian Obirikorang.

**Data curation:** Eric NY Nyarko, Ebenezer Amakye, Emmanuel K Ofori, Michael Appiah, Manfred Anim, Nathaniel Lartey, Samuel Ametepe, Evans Asamoah-Adu, Justice Kumi, Derrick ND Dodoo, Monica Adom, Christian Obirikorang.

**Formal analysis:** Eric NY Nyarko, Ebenezer Amakye, Emmanuel K Ofori, Michael Appiah, Manfred Anim, Nathaniel Lartey, Samuel Ametepe, Evans Asamoah-Adu, Justice Kumi, Derrick ND Dodoo, Monica Adom, Emmanuel Owiredu, Esther Kwafo, Christian Obirikorang.

**Methodology:** Eric NY Nyarko, Ebenezer Amakye, Emmanuel K Ofori, Michael Appiah, Manfred Anim, Nathaniel Lartey, Samuel Ametepe, Evans Asamoah-Adu, Justice Kumi, Derrick ND Dodoo, Monica Adom, Emmanuel Owiredu, Esther Kwafo, Christian Obirikorang.

**Resources:** Eric NY Nyarko, Emmanuel K Ofori, Michael Appiah, Manfred Anim, Nathaniel Lartey, Evans Asamoah-Adu, Justice Kumi, Derrick ND Dodoo, Monica Adom, Christian Obirikorang.

**Software:** Samuel Ametepe, Emmanuel Owiredu, Esther Kwafo.

**Supervision:** Eric NY Nyarko, Ebenezer Amakye, Christian Obirikorang.

**Writing – original draft:** Eric NY Nyarko, Ebenezer Amakye, Christian Obirikorang.

**Writing – review & editing:** Eric NY Nyarko, Ebenezer Amakye, Emmanuel K Ofori, Michael Appiah, Manfred Anim, Nathaniel Lartey, Samuel Ametepe, Evans Asamoah-Adu, Justice Kumi, Derrick ND Dodoo, Monica Adom, Emmanuel Owiredu, Esther Kwafo, Christian Obirikorang.

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
