## [Decision Letter · Decision Letter 0]

19 Dec 2024

PGPH-D-24-02628

Prevalence of H. pylori, S. typhi, P. falciparum, and T. gondii infections and levels of liver function markers among HBV infected Ghanaians: A cross-sectional study in the Greater Accra Region

Dear Dr. Nyarko,

Thank you for submitting your manuscript to PLOS Global Public Health. After careful consideration, we feel that it has merit but does not fully meet PLOS Global Public Health’s publication criteria as it currently stands. Therefore, we invite you to submit a revised version of the manuscript that addresses the points raised during the review process.

We look forward to receiving your revised manuscript.

Kind regards,

Orvalho Augusto, MD, MPH, PhD

Academic Editor

Additional Editor Comments (if provided):

This is an interesting work. The authors aim to determine the prevalence of common bacterial and protozoa parasite infections among patients with active hepatitis B infection. They used a cross-sectional sample of patients with HBV to check for 4 pathogens (H. pylori, S. typhi, P. falciparum, and T. gondi).

Comments/Questions

1. The sample source is not well described. Please provide details for the current descriptions in lines 115 and 116. Is the HBV routine checking in these health facilities? Is there an algorithm to find these patients? Is it something designed just for this study?

2. Why was the sample size computed to detect HBV rather than to estimate with some precision the prevalence of each of the 4 pathogens? A note would be useful as it is too late to amend the sample size anyway.

3. Table 1 is not well formatted for Epidemiology/Public Health use. Please redo the table to have three columns. On for variable, for absolute counts, and for relative frequency. The second column should add up to 120, whereas the third is 100% per variable.

4. Please fill out the STROBE checklist for the cross-sectional form [https://www.strobe-statement.org/checklists/]. Address any item missing in the manuscript.

5. Tables 2 and for the continuous variables:

- OK for the mean and Standard Deviation (SD), and independent t-tests. However, the literature on these variables indicates that they are very skewed. Please show the medians and the IQR as well.

- Please, remove the p-values for HBV vs S. typhi and HBV vs P. falciparum. The Ns are too small.

6. Figure 1 and prevalence estimates:

- please add 95% confidence interval

- Was there any HBV patients with multiple pathogens?

- Figure please sort the bars according to the prevalence.

Reviewers' comments:

Reviewer's Responses to Questions

**Comments to the Author**

1. Does this manuscript meet PLOS Global Public Health’s publication criteria?

Reviewer #1: Yes

2. Has the statistical analysis been performed appropriately and rigorously?

Reviewer #1: Yes

3. Have the authors made all data underlying the findings in their manuscript fully available (please refer to the Data Availability Statement at the start of the manuscript PDF file)?

Reviewer #1: Yes

4. Is the manuscript presented in an intelligible fashion and written in standard English?

Reviewer #1: Yes

Reviewer #1: Reconsider the title; it should not have none-standard abbreviations. Write-out in full HBV and species names i.e., “Prevalence of Helicobacter pylori, Salmonella typhi, Plasmodium falciparum, and Toxoplasma gondii infections and levels of liver function markers among Hepatitis B Virus infected Ghanaians: A cross-sectional study in the Greater Accra Region”

ABSTRACT

Rephrase for clarity; for example, the first sentence could be rephrased as “Coinfection of humans with Hepatitis B Virus (HBV) and non-viral pathogens may worsen the outcome of HBV infection on the liver.”

Define ‘LFM’s’; ALT, etc., at first use of term

You state that H. pylori, S. typhi, P. falciparum, and T. gondii were screened for, from the blood using lateral flow immunochromatographic assays (LFM). Yet, lines 153-155 (Methods section - in the main text) you state that “The P. falciparum positive samples were confirmed by a trained microscopic examination of thin and thick film prepared for blood smear.” Please reconcile.

INTRODUCTION

Lines 70-71: GAR has already been defined but you continue to use the full term (Greater Accra Region) thereafter.

METHODS

Regarding P. falciparum detection, reconcile the information in lines 153-155 with that in lines 41-43 (abstract).

Furthermore, immunological/lateral flow-based assays (LFM) were used to detect all the pathogens under study except P. falciparum that was detected by using microscopy. Yet, LFM assays for detection of P. falciparum also do exist, so I wonder what the rationale of using microscopy, which has a very low sensitivity, was.

Relatedly, for S. typhi, the immunology-based assays need to be interpreted with caution as they usually imply prior exposure, not necessarily recent/active infection and tend to have a very high sensitivity.

RESULTS

No major issues

DISCUSSION

The authors have not discussed the key limitations regarding detection of certain pathogens under study, which have been pointed out in methods.

First, immunological/lateral flow-based assays (LFM) were used to detect all the pathogens under study except P. falciparum that was detected by using microscopy. Yet, LFM assays for P. falciparum detection do exist; I wonder what the rationale of using microscopy, which has a very low sensitivity, was. One is tempted to imagine that the prevalence of P. falciparum could be higher than what was reported if a more sensitive detection method was adopted.

Relatedly, for S. typhi, the immunology-based assays need to be interpreted with caution as they usually imply prior exposure, not necessarily recent/active infection and tend to have a very high sensitivity.

**Do you want your identity to be public for this peer review?** For information about this choice, including consent withdrawal, please see our Privacy Policy

Reviewer #1: **Yes: ** David Patrick Kateete

---

## [Decision Letter · Decision Letter 1]

13 May 2025

PGPH-D-24-02628R1

Prevalence of Helicobacter pylori, Salmonella typhi, Plasmodium falciparum, and Toxoplasma gondii infections and levels of liver function markers among Hepatitis B Virus infected Ghanaians: A cross-sectional study in the Greater Accra Region

Dear Dr. Nyarko,

Thank you for submitting your manuscript to PLOS Global Public Health. After careful consideration, we feel that it has merit but does not fully meet PLOS Global Public Health’s publication criteria as it currently stands. Therefore, we invite you to submit a revised version of the manuscript that addresses the points raised during the review process.

We look forward to receiving your revised manuscript.

Kind regards,

Miquel Vall-llosera Camps

Staff Editor

Additional Editor Comments:

It was considered necessary to invite additional reviewers after the previous reviewer declined. One of the reviewers is more positive about your study, but the other reviewer has raised remaining concerns that need to be addressed.

Reviewers' comments:

Reviewer's Responses to Questions

**Comments to the Author**

Reviewer #2: (No Response)

Reviewer #3: (No Response)

publication criteria?

Reviewer #2: Yes

Reviewer #3: Yes

3. Has the statistical analysis been performed appropriately and rigorously?

Reviewer #2: Yes

Reviewer #3: Yes

4. Have the authors made all data underlying the findings in their manuscript fully available (please refer to the Data Availability Statement at the start of the manuscript PDF file)?

Reviewer #2: Yes

Reviewer #3: Yes

5. Is the manuscript presented in an intelligible fashion and written in standard English?

Reviewer #2: Yes

Reviewer #3: Yes

Reviewer #2: This study attempts to provide an overview regarding the prevalence of Heliobacter pylori, Salmonella typhi, Plasmodium falciparum, and Toxoplasma gondii among HBV-infected persons in the Greater Accra Region of Ghana. In addition, the levels of liver function markers were determined in the coinfected patients. The prevalence of H. pylori and T. gondii was high among HBV-infected patients. Liver function tests were elevated in coinfected cases. This study well describes the current status (prevalence) of these infections in HBV-infected patients, and the effects of these coinfections on liver function tests. The findings of this study contribute further evidence for informing reforms to the national guidelines for the management, prevention, and control of these coinfections in HBV positive patients.

Reviewer #3: This study is to determine the Prevalence of Helicobacter pylori, Salmonella typhi, Plasmodium falciparum, and Toxoplasma gondii infections and levels of liver function markers among Hepatitis B Virus infected Ghanaians.

I have some few comments about this study

1. The authors tested for antibodies to certain infections in asymptomatic patients with viral hepatitis. However, the presence of antibodies does not necessarily indicate a current infection, particularly for H. pylori, which has a high prevalence in Ghana (44-70%). Furthermore, the typhoid antibody test (IgM) has a low specificity of approximately 50%, leading to a high rate of false positive results, especially in asymptomatic patients. Therefore, the authors cannot conclusively determine that these organisms were present in the patients at the time of sampling.

2. The authors also found a correlation between high ALT levels and positive H. pylori antibody tests. However, the average ALT levels were within the normal range, suggesting that there was no significant inflammation in these patients. Therefore, it cannot be concluded that H. pylori infection contributes to increased inflammation or accelerates liver disease progression in these patients.

3. The study's findings have limited implications for the monitoring or management of viral hepatitis B in Ghana. The authors should also note that any form of infection can cause inflammation of the liver but tends to be acute without causing chronic infection

4. The results would have been more meaningful if the authors had compared patients with these infections and hepatitis B-associated liver cirrhosis or hepatocellular carcinoma to those with hepatitis B without end-stage liver disease, allowing for a more informative analysis of potential correlations and disease progression.

**Do you want your identity to be public for this peer review?** For information about this choice, including consent withdrawal, please see our Privacy Policy

Reviewer #2: **Yes: ** Fatemeh Farshadpour

Reviewer #3: No

---

## [Decision Letter · Decision Letter 2]

7 Jul 2025

PGPH-D-24-02628R2

Prevalence of Helicobacter pylori, Salmonella typhi, Plasmodium falciparum, and Toxoplasma gondii infections and levels of liver function markers among Hepatitis B Virus infected Ghanaians: A cross-sectional study in the Greater Accra Region

Dear Dr. Nyarko,

Thank you for submitting your manuscript to PLOS Global Public Health. After careful consideration, we feel that it has merit but does not fully meet PLOS Global Public Health’s publication criteria as it currently stands. Therefore, we invite you to submit a revised version of the manuscript that addresses the points raised during the review process.

Please note the additional question raised by Reviewer #2. I have also added my comments below based on my own review.

We look forward to receiving your revised manuscript.

Kind regards,

Sanghyuk S Shin

Academic Editor

Journal Requirements:

Additional Editor Comments (if provided):

- Please make sure that the manuscript aligns with the reporting recommendations for cross-sectional studies as found in STROBE: https://www.strobe-statement.org/.

- In the Abstract and Results, please report the 95% confidence intervals for the prevalence estimates for the 4 co-infections.

- In the Abstract, the statement about elevated markers for S. typhi and P. falciparum co-infected participants is misleading, as the differences are not statistically significant and the number of S. typhi and P. falciparum infections are very small. Given the lack of statistical evidence of differences, please remove these results from the abstract and make sure that the Results and Discussion reflect lack of statistically significant differences for these comparisons.

- Please add a clear sentence describing the objective of the study in the last paragraph of the Introduction. Currently, it states "Hence this study".

Reviewers' comments:

Reviewer's Responses to Questions

**Comments to the Author**

Reviewer #2: All comments have been addressed

Reviewer #3: (No Response)

publication criteria?

Reviewer #2: Yes

Reviewer #3: Yes

3. Has the statistical analysis been performed appropriately and rigorously?

Reviewer #2: Yes

Reviewer #3: Yes

4. Have the authors made all data underlying the findings in their manuscript fully available (please refer to the Data Availability Statement at the start of the manuscript PDF file)?

Reviewer #2: Yes

Reviewer #3: Yes

5. Is the manuscript presented in an intelligible fashion and written in standard English?

Reviewer #2: Yes

Reviewer #3: Yes

Reviewer #2: This study attempts to provide an overview regarding the prevalence of Heliobacter pylori, Salmonella typhi, Plasmodium falciparum, and Toxoplasma gondii among HBV-infected persons in the Greater Accra Region of Ghana. In addition, the levels of liver function markers were determined in the coinfected patients. The prevalence of H. pylori and T. gondii was high among HBV-infected patients. Liver function tests were elevated in coinfected cases. This study well describes the current status (prevalence) of these infections in HBV-infected patients, and the effects of these coinfections on liver function tests. The findings of this study contribute further evidence for informing reforms to the national guidelines for the management, prevention, and control of these coinfections in HBV-positive patients.

Reviewer #3: The authors response well noted.

I still belief that using antibodies of the co-infections investigated such as the H. pylori antibodies does not indicate current infection especially in endemic countries like Ghana and currently not useful to diagnosed or to confirm eradication rate. Though previous studies show an association of H. pylori infection and HCC in patients with risk factors of HCC, but using blood samples to determine H. pylori Ab irrespective of the specificity and sensitivity does no confirm the presence of H. pylori

One of the current recommendations for managing hepatitis B is persistently high ALT, that is ALT above the reference range. Though there was a statistically significant association between ALT and H. pylori, the average ALT for those with H. pylori, was it above the reference range of the laboratory values used?

**Do you want your identity to be public for this peer review?** For information about this choice, including consent withdrawal, please see our Privacy Policy

Reviewer #2: **Yes: ** Fatemeh Farshadpour

Reviewer #3: No

---

## [Editor Report · Decision Letter 3]

8 Aug 2025

Prevalence of Helicobacter pylori, Salmonella typhi, Plasmodium falciparum, and Toxoplasma gondii infections and levels of liver function markers among Hepatitis B Virus infected Ghanaians: A cross-sectional study in the Greater Accra Region

PGPH-D-24-02628R3

Dear Dr Nyarko,

We are pleased to inform you that your manuscript 'Prevalence of Helicobacter pylori, Salmonella typhi, Plasmodium falciparum, and Toxoplasma gondii infections and levels of liver function markers among Hepatitis B Virus infected Ghanaians: A cross-sectional study in the Greater Accra Region' has been provisionally accepted for publication in PLOS Global Public Health.

Best regards,

Sanghyuk S Shin

Academic Editor